# Early Mortality among Patients with Head and Neck Cancer Diagnosed in Thuringia, Germany, between 1996 and 2016—A Population-Based Study

**DOI:** 10.3390/cancers14133099

**Published:** 2022-06-24

**Authors:** Mussab Kouka, Jens Buentzel, Holger Kaftan, Daniel Boeger, Andreas H. Mueller, Andrea Wittig, Stefan Schultze-Mosgau, Thomas Ernst, Orlando Guntinas-Lichius

**Affiliations:** 1Department of Otorhinolaryngology, Jena University Hospital, 07747 Jena, Germany; mussab.kouka@med.uni-jena.de; 2Department of Otorhinolaryngology, Suedharzklinikum Nordhausen, 99734 Nordhausen, Germany; jens.buentzel@shk-ndh.de; 3Department of Otorhinolaryngology, Helios-Klinikum Erfurt, 99089 Erfurt, Germany; holger.kaftan@helios-gesundheit.de; 4Department of Otorhinolaryngology, SRH Zentralklinikum Suhl, 98527 Suhl, Germany; daniel.boeger@srh.de; 5Department of Otorhinolaryngology, SRH Wald-Klinikum Gera, 07548 Gera, Germany; andreas.mueller@srh.de; 6Department of Radiotherapy and Radiation Oncology, Jena University Hospital, 07743 Jena, Germany; andrea.wittig-sauerwein@med.uni-jena.de; 7Department of Oromaxillofacial Surgery and Plastic Surgery, Jena University Hospital, 07747 Jena, Germany; stefan.schultze-mosgau@med.uni-jena.de; 8University Tumor Center, Jena University Hospital, 07747 Jena, Germany; thomas.ernst@med.uni-jena.de

**Keywords:** early mortality, 30-day mortality, 90-day mortality, 180-day mortality, early death, head and neck cancer, survival

## Abstract

**Simple Summary:**

When we consider the outcome of cancer treatment, we mostly focus on overall survival: studies on early mortality in head and neck cancer (HNC) are sparse. This retrospective population-based study investigated early mortality of HNC and the influence of patients’ tumor and treatment characteristics. All 8288 patients with primary HNC of the German federal state Thuringia from 1996 to 2016 were included. Statistics were performed to identify independent factors for 30-day, 90-day, and 180-day mortality. The 30-, 90-, and 180-day mortality risks were 1.8%, 5.1%, and 9.6%, respectively. Male sex, increasing age, larger tumor size, and distant metastasis, tumors of the cavity of mouth, oropharynx, and hypopharynx had a significantly greater 180-day mortality. Surgery, radiotherapy, and multimodal therapy were associated with decreased 180-day mortality. Typical factors associated with worse overall survival had the most important impact on early mortality in HNC patients in a population-based setting.

**Abstract:**

Population-based studies on early mortality in head and neck cancer (HNC) are sparse. This retrospective population-based study investigated early mortality of HNC and the influence of patients’ tumor and treatment characteristics. All 8288 patients with primary HNC of the German federal state Thuringia from 1996 to 2016 were included. Univariate and multivariate analysis were performed to identify independent factors for 30-day, 90-day, and 180-day mortality. The 30-, 90-, and 180-day mortality risks were 1.8%, 5.1%, and 9.6%, respectively. In multivariable analysis, male sex (odds ratio (OR) 1.41; 95% confidence interval (CI) 1.08–1.84), increasing age (OR 1.81; CI 1.49–2.19), higher T (T4: OR 3.09; CI 1.96–4.88) and M1 classification (OR 1.97; CI 1.43–2.73), advanced stage (IV: OR 3.97; CI 1.97–8.00), tumors of the cavity of mouth (OR 3.47; CI 1.23–9.75), oropharynx (OR 3.01; CI 1.06–8.51), and hypopharynx (OR 3.27; CI 1.14–9.40) had a significantly greater 180-day mortality. Surgery (OR 0.51; CI 0.36–0.73), radiotherapy (OR 0.37; CI 0.25–0.53), and multimodal therapy (OR 0.10; CI 0.07–0.13) were associated with decreased 180-day mortality. Typical factors associated with worse overall survival had the most important impact on early mortality in a population-based setting.

## 1. Introduction

Germany has a population of approximately 83 million inhabitants [1]. In 2018, the overall number of new cancer cases was 497,885. The number of new head and neck cancer (HNC) cases was 14,312, which represents 2.8% of all cancers [2]. Tumor staging is performed according to the TNM classification and the UICC (Union for international Cancer Control) system [3]. HNC is the seventh most common cancer worldwide and is often diagnosed at an advanced stage (stage III/IV), which is associated with a lower overall survival (OS) [4,5]. Approximately 34% of the patients die within 5 years after diagnosis [6,7]. Death of the patients can also occur early after diagnosis and during the period of treatment. Early mortality in HNC in this regard is poorly studied. Early mortality describes premature death within a defined interval (e.g., 30-day, 90-day, or 180-day mortality) after the date of diagnosis.

In nationwide studies of the Swedish and Finnish databases, early mortality at 180 days was about 9–10% [8]. The causes of early mortality may be either tumor-related, patient-related, treatment-related, or due to perioperative mortality. Few studies have examined early mortality in HNC patients in an epidemiologic population-based setting [9,10,11]. Based on data from the Danish Head and Neck Cancer Group (DAHANCA), 180-day mortality was 7.1% for all HNC patients treated with curative radio(chemo)therapy in Denmark between 2000 and 2017 [12]. In the Swedish Head and Neck Cancer Registry (SweHNCR) population-based study between 2008 and 2015, 180-day early mortality rate of 9733 HNC patients was 9.5% [9]. A retrospective analysis of the U.S. National Cancer Data Base (NCDB) of HNC cases between 2004 and 2014 examined demographic and socioeconomic factors that influence early mortality. Gender, race/ethnicity, comorbidity, other socioeconomic factors, interval between diagnosis to treatment, income, and insurance were associated with decreased 90-day mortality [10]. 

In general, most evidence comes from larger monocentric series and retrospective studies and therefore has methodologic limitations. Therefore, a population-based analysis of cancer registry data of all new Thuringian patients treated for primary HNC between 1996 and 2016 was performed. The main task was to investigate the influence of patients’ characteristics, tumor characteristics, and treatment types on early mortality at 30, 90, and 180 days after initial diagnosis.

## 2. Material and Methods

### 2.1. Ethical Considerations

This retrospective study was approved by the Ethics Committee of the Jena University Hospital (IRB No. 3204-07/11). The Ethics Committee waived the requirement for informed consent of the patients because the study had a non-interventional retrospective design and all data were analyzed anonymously.

### 2.2. Patients

This population-based cohort study was conducted using patient data from the five Thuringian cancer registries (Gera, Erfurt, Nordhausen, Suhl, Weimar, and Jena). All patients who were diagnosed with primary HNC in the Thuringian cancer registries from January 1996 to December 2016 were included. Thuringia is one of 16 states in the Federal Republic of Germany and has a population of 2.2 million people. All patients who had skin cancer or who had metastases from other entities in the head and neck region were excluded. Duplicates were also excluded, i.e., in patients with multiple head and neck cancers, only the index cancer was counted. A total of 8288 new cases of primary head and neck cancer were registered. The cases were divided into salivary glands, lip, oral cavity, oropharynx, nasopharynx, hypopharynx, larynx, nasal/paranasal cavity, middle ear, and unspecified according to the International Classification of Diseases for Oncology (ICD-O-3) [13]. The clinical (or if the patient underwent primary surgery) pathological T, N, and M categories were recorded and grouped together to the UICC tumor stage of the primary tumor using the TNM classification, 7th edition [7]. If a case was originally classified in accordance with an older edition, the case was reclassified according to the 7th edition. To allow an optimal comparison with other population-based studies, the tumors were additionally categorized according to the Surveillance, Epidemiology, and End Results Program (SEER) classification system into primary tumors without metastasis (localized), tumor with cervical metastasis (regional), and tumor with distal metastasis (distal).

### 2.3. Statistical Analysis

SPSS Statistics version 25 (IBM Deutschland GmbH, 71139 Ehningen, Germany) was used to perform the statistical analyses. The 30-day, 90-day, and 180-day mortality, respectively, was defined as any patient who died within these three periods. Nominal or ordinal parameters for patients’ tumor and treatment characteristics were compared between the patients who were alive and died within the three periods using chi-squared and Fisher exact tests. Significant factors from these univariate analyses were included into multivariate binary logistic regression models (death: yes/no) to identify independent factors separately for 30-day, 90-day, and 180-day mortality, respectively. Missing data were minimal (<5%) for outcomes and predictor variables employed in the reported analyses, and hence no imputation was employed. Unadjusted odds ratios (OR) are reported with 95% confidence intervals (CI). All statistical tests were 2-sided. The significance level of *p* = 0.05 was set. 

## 3. Results

### 3.1. Patients’ Characteristics, Tumor Characteristics, and Treatment Characteristics

The distribution of patient characteristics, tumor characteristics, and treatment characteristics factors in the total cohort and in the subgroups with 30-day, 90-day, and 180-day mortality is shown in Table 1. The median age at diagnosis was 60 years and men formed the majority of HNC patients (6540 men, 78.9%). Most patients had T1-2 tumors, 40% had N0 status, and 87% had no evidence of distant metastases. Most tumors were located in the larynx (21%), pharynx (40.7%), and oral cavity (27%). Of the total group of 8288 HNC patients, 151 patients (1.8%) died within 30 days, 426 patients (5.1%) within 90 days, and 798 patients (9.6%) within 180 days after diagnosis of HNC. Mortality rates related to gender, age, cancer staging, and treatment are summarized in Figure 1. The mortality at 30, 90, and 180 days was higher in male patients and in patients older than the median age. Mortality increased with higher tumor stage, especially for stage IV cancer. If no curative therapy was possible, about one third of the patients died within 90 days, and nearly half of the patients within 180 days after diagnosis.

### 3.2. Univariate Analysis of Predictors for Early Mortality

There was a significant survival difference in sex and age in all inclusion periods (Table 1). Male sex (death within 30 days: 85.4%, *p* = 0.047; death within 90 days: 84.3%, *p* = 0.005; death within 180 days: 82.8%, *p* = 0.004) and higher age (death within 30 days: 73.5%, death within 90 days: 70.9%, and death within 180 days: 64.5%; all *p* < 0.001) was associated with higher early mortality probability. Higher T-, N-, and M-classification and advanced cancer stage showed significant influence for all inclusion periods of 30-day, 90-day, and 180-day mortality (all *p* < 0.001). Tumor localization showed no significant impact on the 30-day mortality (*p* = 0.084). The probability of 90-day and 180-day mortality varied depending on the primary tumor localization (*p* < 0.001). The worst prognosis was found for tumors located in the cavity of the mouth (25.9%), oropharyngeal (32.6%), and hypopharyngeal carcinoma (17.5%) with death occurring within 180 days (*p* < 0.001). 

Any treatment type also showed a significant influence on early mortality for all inclusion periods (all *p* < 0.001). Patients who underwent radiotherapy (death within 30 days: 7.3%; death within 90 days 13.8%; death within 180 days: 16.5%), surgery (death within 30 days: 39.1%; death within 90 days: 30.8%; death within 180 days: 24.9%) or multimodal therapy (death within 30 days: 8.6%; death within 90 days: 23.2%; death within 180 days: 35.6%) had lower risk for early mortality (all *p* < 0.001). In summary, patients’ tumor and treatment related factors all influenced the risk of death within 30 days, 90 days, and 180 days except tumor localization in the 30-day mortality (*p* = 0.084). 

### 3.3. Multivariate Analysis of Predictors for Early Mortality

The odds ratio (OR) and 95% confidence interval (CI) for 30-day, 90-day, and 180-day mortality are shown in Table 2. In the multivariable logistic regression analysis, all variables which showed significance in the univariate analysis were included. Men had a 3.8-fold increased odds ratio than women (OR 3.81; 95% CI 1.71–8.49; *p* = 0.001) for 30-day mortality, 1.8-fold increased odds ratio for 90-day mortality (OR 1.81; 95% CI 1.23–2.65; *p* = 0.003), and 1.4-fold increased OR for 180-day mortality (OR 1.41; 95% CI 1.080–1.836; *p* = 0.012). Increased age at diagnosis (30-day mortality: OR 2.137; 95% CI 1.33–3.43; *p* = 0.002; 90-day mortality: OR 2.16; 95% CI 1.64–2.84; *p* < 0.001; 180-day mortality: OR 1.81; 95% CI 1.49–2.19; *p* < 0.001) and recurrence (30-day mortality: OR 0.11; 95% CI 0.02–0.77; *p* = 0.027; 90-day mortality: OR 0.12; 95% CI 0.05–0.29; *p* < 0.001; 180-day mortality: OR 0.46; 95% CI 0.32–0.63; *p* < 0.001) showed to be independent variables for all inclusion periods of 30-day, 90-day, and 180-day mortality. All treatment types except for surgery in the 30-day mortality showed also a significant influence for all inclusion periods. Surgery (90-day mortality: OR 0.55; 95% CI 0.37–0.82; *p* < 0.001; 180-day mortality: OR 0.51; 95% CI 0.36–0.73; *p* < 0.001), radiotherapy (90-day mortality: OR 0.19; 95% CI 0.12–0.31; *p* < 0.001; 180-day mortality: OR 0.37; 95% CI 0.25–0.53; *p* < 0.001) and multimodal therapy (90-day mortality: OR 0.05; 95% CI 0.04–0.08; *p* < 0.001; 180-day mortality: OR 0.10; 95% CI 0.07–0.13; *p* < 0.001) were associated with decreased 90-day and 180-day mortality.

Radiotherapy (OR 0.10; 95% CI 0.05–0.23; *p* < 0.001) and multimodal therapy (OR 0.02; 95% CI 0.01–0.03; *p* < 0.001) showed decreased 30-day mortality. 

The expected important predictors such as T3 (90-day mortality: OR 2.45; 95% CI 1.19–5.05; *p* = 0.015; 180-day mortality: OR 1.88; 95% CI 1.17–3.01; *p* = 0.009) and T4 classification (90-day mortality: OR 4.15; 95% CI 2.040–8.46; *p* < 0.001; 180-day mortality: OR 3.09; 95% CI 1.96–4.88; *p* < 0.001), and cancer stage II (90-day mortality: OR 3.37; 95% CI 1.23–9.22; *p* = 0.018; 180-day mortality: OR 3.51; 95% CI 1.56–7.91; *p* = 0.002), stage III (90-day mortality: OR 3.18; 95% CI 1.01–10.00; *p* = 0.048; 180-day mortality: OR 2.98; 95% CI 1.28–6.94; *p* = 0.012), and stage IV (90-day mortality: OR 3.07; 95% CI 1.01–9.32; *p* = 0.048; 180-day mortality: OR 3.97; 95% CI 1.97–8.00; *p* < 0.001) were associated with an increased risk of 90-day and 180-day mortality. There was no significant influence for all inclusion periods by higher N-classification except for N3–classification during 30-day mortality (OR: 11.99; 95% CI 1.23-110.65; *p* = 0.028). In 180-day mortality, M1 classification (OR 1.97; 95% CI 1.43–2.73; *p* < 0.001), tumors of the cavity of the mouth subsite (OR 3.47; 95% CI 1.23–9.75; *p* = 0.019), oropharynx (OR 3.01; 95% CI 1.06–8.51; *p* = 0.038), and hypopharynx (OR 3.27; 95% CI 1.14–9.40; *p* = 0.028) additionally showed significant influence. In total, odds ratios for 90-day and 180-day mortality were similar for most variables. There were less significant independent factors for 30-day mortality than for 90-day and 180-day mortality. Mortality risk for higher age, recurrence, and treatment types of 30-day mortality were similar to that of 90-day and 180-day mortality. The odds ratio for male sex during 30-day mortality was 3.819, indicating that men have more than twice the risk of dying within 30 days than within 90 days (OR: 1.81; 95% CI 1.23–2.65; *p* = 0.003) or 180 days (OR: 1.41; 95% CI 1.08–1.84; *p* = 0.012). 

## 4. Discussion

Early mortality of HNC patients in terms of incidence and influencing factors has been rarely investigated worldwide. In Germany, early mortality of HNC patients has not been analyzed so far. To our knowledge, this study provides for the first time a population-based analysis of early mortality of HNC patients in Germany. The aim was to investigate the role of premature mortality and the influence of patients’ characteristics, tumor characteristics, and treatment types. The odds ratios for higher age, recurrence, and treatment types of 30-day mortality were similar to those for 90-day and 180-day mortality whereas the odds ratio for men of 30-day mortality was twice that during 90-day and 180-day mortality. In total, the odds ratios for 90-day and 180-day mortality were similar for most variables. 

Two previous Swedish studies based on data from 6785 and 9733 HNC patients reported 180-day mortality of 9.8% and 9.5%, respectively [7,8]. These results are consistent with our 180-day mortality of 9.6%. Jensen et al. reported a 90-day and 180-day mortality of 3.1% and 7.1% based on data from the Danish Head and Neck Cancer Group (DAHANCA) for 11,419 HNC patients treated with curative radio(chemo)therapy [12]. However, a comparison of treatment characteristic should be considered. The decreased risk in the Danish study is associated with HNC patients treated with primary radio(chemo)therapy [13]. In the present study, all types of therapy were considered. Otherwise, the present dataset would have been subject to an uncontrolled bias here. Therefore, the differences in early mortality may be influenced by the different treatment types and less investigated factors in the Danish study [14]. In randomized, multicenter clinical trials for treatment of HNC, 30-day mortality has been about 1.5−3.3% [15,16,17]. These results are consistent with the 30-day mortality rate of 1.8% in the present study. Most studies on early mortality of HNC examined only one period of early mortality (often 90-day mortality or 180-day mortality) [8,10,11,18]. Only a few studies observed more than one period of early mortality [9,12]. This study is the first to compare three inclusion periods of early mortality.

There was no clear evidence of a period with a higher risk of early mortality for HNC patients, i.e., there was no relevant difference between 30, 90, and 180 days after diagnosis of HNC. The risk of early death was related to patients’ tumor and treatment characteristics. Increasing age and male gender were associated with a significantly increased risk of early mortality. Broadly consistent with previous studies were increasing age, advanced T-classification, increased stage, the location of the hypopharyngeal tumor and oral cavity carcinoma, which were associated with an increased risk of early mortality [9,11,12,18,19,20]. In general, the factors associated with early mortality are the same factors associated with worse overall survival of HNC [21]. However, important factors influencing overall survival such as alcohol consumption, smoking habits, and HPV status were not considered in this study [22]. 

There was a significant decrease of odds ratios of the investigated factors for most variables during the inclusion periods of 30-day, 90-day, and 180-day mortality. We suggest that the decreasing mortality risk is associated with increasing advances in imaging diagnostics and treatment decision-making, including multidisciplinary tumor boards and generally improved medical and nursing treatment care during the time after diagnosis of HNC.

However, this study has some limitations. The retrospective design cannot guarantee sufficient information and standardized treatment decisions. Additionally, the actual causes of death were not reported. It is not clear whether death was caused by the cancer itself, by treatment-related complications, or by concomitant diseases. In addition, insufficient patient data and lack of information on alcohol consumption, smoking habits, occupational exposures, oral hygiene, and HPV status are limiting factors in this study [23]. Socioeconomic factors, comorbidity, and frailty were also not taken into account in this study, although there is evidence of a significant influence on early mortality [10,18,24]. Gaubatz et al. demonstrated the significant influence of socioeconomic factors such as gender, race/ethnicity, comorbidity, distance to treatment, income, and insurance in a retrospective analysis of the U.S. National Cancer Data Base (NCDB) of 260,011 HNC patients [10]. However, this study, with its large patient population, homogeneity, and population-based nature, may offset some of the limiting factors. We demonstrated that increasing age, male sex, higher T (T3, T4) and M1 classification, advanced stage, cavity of mouth subsite, oropharyngeal and hypopharyngeal carcinoma were significant factors for 180-day mortality. In addition, some of our results were confirmed in a previous study on the same patient population. Dittberner et al. showed that male gender, higher tumor stage, and hypopharyngeal carcinoma were also independent factors with a negative impact on overall survival in the same cohort of patients [25]. The surgical techniques, the modalities of radiotherapy, and the concepts of multimodal therapy have changed between 1996 and 2016. For instance, the introduction of transoral robotic surgery might have decreased early mortality [26]. The influence of innovative surgical techniques and other therapy changes on early mortality were not analyzed. Immunotherapy has opened a new era in cancer treatment and was introduced for head and neck cancer in 2018, hence later than the study period analyzed in the present trial [27]. It remains to be seen if immunotherapy, when it has reached primary therapy concepts [28], will help to decrease early mortality. Since the legal regulations and responsibilities for cancer registration in Germany have changed within 20 years, bias and underreporting of incident cases cannot be excluded. Nevertheless, the Thuringian cancer registries have probably covered about 98% of all head and neck cancer patients in Thuringia over the period of the study [29,30].

The results in this study and additional clinical information can be used to identify patients with high risk of early mortality. Curative and palliative treatments could be optimized for HNC patients with high risk of early mortality. In this study, almost every type of treatment was associated with decreased risk for early death during all inclusion periods of 30-day, 90-day and 180-day mortality. Therefore, it is important to reduce treatment delays to prevent early death. In summary, the 30-day, 90-day, and 180-day mortality risks were 1.8%, 5.1%, and 9.6%, respectively. The odds ratios for higher age, recurrence, and treatment types were similar to all inclusion periods of 30-day, 90-day, and 180-day mortality. The odds ratios for male sex, T-classification, and cancer stage were very similar for 90-day and 180-day mortality. Further studies with more attention to relevant influencing factors with the potential to reduce early death need to be performed. 

## 5. Conclusions

This study provides for the first time a population-based analysis of 8288 HNC patients in Germany investigating early mortality 30 days, 90 days, and 180 days after diagnosis of HNC between 1996 and 2016. In summary, the 30-day, 90-day, and 180-day mortality risks were 1.8%, 5.1%, and 9.6%, respectively. Increased age at diagnosis, male sex, recurrence, and treatment types showed to be independent factors for all inclusion periods of 30-day, 90-day, and 180-day mortality. For 90-day and 180-day mortality, advanced T classification, and advanced stage were significant factors. Only in 180-day mortality tumors of cavity of mouth, oropharynx, and hypopharynx were additionally found to be independent factors. The mortality risks for higher age, recurrence, and treatment types were similar for the 30-day, 90-day, and 180-day mortality. In contrast, the mortality risks for male sex, T-classification, and cancer stage were only very similar for the 90-day and 180-day mortality. For 30-day mortality, only mortality risk for male sex was higher than for 90-day and 180-day mortality. Further studies with more attention to relevant influencing factors with the potential to reduce early death need to be performed.

## Figures and Tables

**Figure 1 cancers-14-03099-f001:**
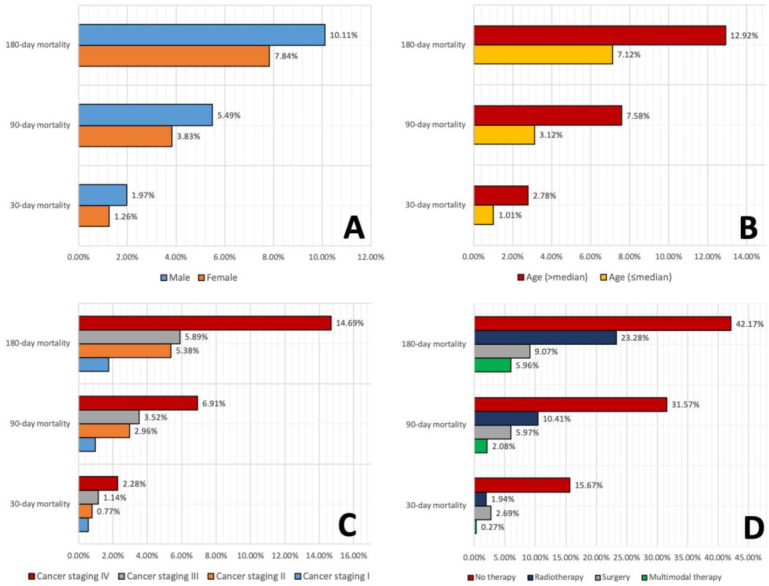
Distribution of 30-day, 90-day, and 180-day mortality according to gender (**A**), age in relation to the median of 60 years (**B**), cancer staging (**C**), and treatment (**D**).

**Table 1 cancers-14-03099-t001:** Univariable analyses on predictors for 30-day, 90-day, and 180-day mortality.

	All		Death within 30 days		*p*	Death within 90 days		*p*	Death within 180 days		*p* *
**Parameter**	** *N* **	**%**	** *N* **	**%**		** *N* **	**%**		** *N* **	**%**	
**Overall**	8288	100.0	151	100.0		426	100.0		798	100.0	
**Gender**					**0.047**			**0.005**			**0.004**
Female	1748	21.1	22	14.6		67	15.7		137	17.2	
Male	6540	78.9	129	85.4		359	84.3		661	82.8	
**Age**					**<0.001**			**<0.001**			**<0.001**
≤Median	3974	47.9	40	26.5		124	29.1		283	35.5	
>Median	3986	48.1	111	73.5		302	70.9		515	64.5	
Unknown	328	4.0	0	0		0	0		0	0	
**T classification**					**<0.001**			**<0.001**			**<0.001**
T1	2019	24.4	13	8.6		31	7.3		60	7.5	
T2	1836	22.2	20	13.2		56	13.1		114	14.3	
T3	1303	15.7	16	10.6		68	16.0		132	16.5	
T4	1941	23.4	60	39.7		172	40.4		351	44.0	
Tx	1189	14.3	41	27.2		99	23.2		141	17.7	
**N classification**					**<0.001**			**<0.001**			**<0.001**
N0	3352	40.4	35	23.1		92	21.6		170	21.3	
N1	783	9.5	8	5.3		36	8.5		61	7.6	
N2	2457	29.6	45	29.8		143	33.6		320	40.1	
N3	285	3.4	16	10.6		36	8.5		74	9.3	
Nx	1411	17.1	47	31.1		119	27.9		173	21.7	
**M classification**					**<0.001**			**<0.001**			**<0.001**
M0	87.0	87.0	108	71.5		321	75.4		615	77.1	
M1	362	4.4	18	11.9		49	11.5		108	13.5	
Mx	714	8.6	25	16.6		56	13.1		75	9.4	
**Cancer staging**					**<0.001**			**<0.001**			**<0.001**
Stage I	1435	17.3	8	5.3		14	3.3		25	3.1	
Stage II	911	11.0	7	4.6		27	6.3		49	6.1	
Stage III	1052	12.7	12	7.9		37	8.7		62	7.8	
Stage IV	3546	42.8	81	53.6		245	57.5		521	65.3	
Unknown	1344	16.2	43	28.5		103	24.2		141	17.7	
**SEER staging**					**<0.001**			**<0.001**			**<0.001**
Localized	3294	39.7	34	22.5		85	20.0		158	19.8	
Regional/distal	3636	43.9	72	47.7		234	54.9		493	61.8	
Unstaged	1358	16.4	45	29.8		107	25.1		147	18.4	
**Localization**					**0.084**			**<0.001**			**<0.001**
Lip	275	3.3	1	0.7		4	0.9		7	0.9	
Cavity of the mouth	2116	25.5	47	31.1		110	25.8		207	25.9	
Oropharynx	2240	27.0	48	31.8		140	32.9		260	32.6	
Nasopharynx	191	2.3	2	1.3		8	1.9		22	2.8	
Hypopharynx	941	11.4	22	14.6		70	16.4		140	17.5	
Larynx	1737	21.0	22	14.6		57	13.4		100	12.5	
Nose and paranasal sinus	246	3.0	5	3.3		16	3.8		23	2.9	
Middle ear	7	0.1	0	0.0		0	0.0		1	0.1	
Salivary glands	490	5.9	4	2.6		21	4.9		38	4.8	
Unspecified	45	0.5	0	0.0		0	0.0		0	0.0	
**Recurrence**					**<0.001**			**<0.001**			**<0.001**
Yes	1330	16.0	5	3.3		13	3.1		55	6.9	
No	6958	84.0	146	96.7		413	96.9		743	93.1	
**Treatment**					**<0.001**			**<0.001**			**<0.001**
No therapy	434	5.2	68	45.0		137	32.2		183	22.9	
Radiotherapy alone	567	6.8	11	7.3		59	13.8		132	16.5	
Surgery alone	2193	26.5	59	39.1		131	30.8		199	24.9	
Multimodal therapy	4766	57.5	13	8.6		99	23.2		284	35.6	
Unknown	328	4.0	NA	NA		NA	NA		NA	NA	

* significant values (*p* < 0.05) in bold; NA = not applicable; SEER = Surveillance, Epidemiology, and End Results Program.

**Table 2 cancers-14-03099-t002:** Multivariate analyses on predictors for 30-day, 90-day, and 180-day mortality.

		30-Day Mortality				90-Day Mortality				180-Day Mortality			
Parameter		OR ***	Lower 95% CI	Upper 95% CI	*p*	OR ***	Lower 95% CI	Upper 95% CI	*p*	OR ***	Lower 95% CI	Upper 95% CI	*p ***
Gender	Female	1	Reference			1	Reference			1	Reference		
	Male	3.81	1.71	8.49	**0.001**	1.81	1.23	2.65	**0.003**	1.41	1.08	1.84	**0.012**
Age	≤Median	1	Reference			1	Reference			1	Reference		
	>Median	2.14	1.33	3.43	**0.002**	2.16	1.64	2.84	**<0.001**	1.81	1.49	2.19	**<0.001**
T classification	T1	1	Reference			1	Reference			1	Reference		
	T2	2.10	0.56	7.848	0.269	1.42	0.66	3.03	0.371	1.14	0.69	1.87	0.608
	T3	1.13	0.29	4.38	0.857	2.45	1.19	5.05	**0.015**	1.88	1.17	3.01	**0.009**
	T4	3.98	1.11	14.32	0.034	4.15	2.04	8.46	**<0.001**	3.09	1.96	4.88	**<0.001**
N classification	N0	1.00	Reference			1	Reference			1	Reference		
	N1	2.68	0.26	27.53	0.408	0.97	0.306	3.09	0.962	0.83	0.34	2.03	0.677
	N2	5.32	0.61	46.14	0.130	0.96	0.329	2.81	0.943	1.12	0.49	2.60	0.789
	N3	11.99	1.30	110.65	**0.028**	1.74	0.57	5.36	0.335	2.13	0.89	5.12	0.091
M classification	M0	1.00	Reference			1	Reference			1	Reference		
	M1	1.71	0.88	3.35	0.115	1.36	0.867	2.13	0.180	1.97	1.43	2.73	**<0.001**
UICC staging	I	1.00	Reference			1	Reference			1	Reference		
	II	1.16	0.22	6.15	0.862	3.37	1.23	9.22	**0.018**	3.51	1.56	7.91	**0.002**
	III	5.82	0.91	37.41	0.064	3.18	1.01	10.00	**0.048**	2.98	1.28	6.94	**0.012**
	IV	3.15	0.55	17.97	0.196	3.07	1.01	9.32	**0.048**	3.97	1.97	8.00	**0.001**
UICC staging categorized	I/II	1.00	Reference			1	Reference			1	Reference		
	III/IV	1.08	0.35	3.34	0.894	1.03	0.49	2.15	0.936	1.29	0.72	2.30	0.389
SEER staging	Localized	1.00	Reference			1	Reference			1	Reference		
	Regional/distal	0.29	0.03	2.79	0.286	1.93	0.62	6.01	0.259	1.57	0.65	3.80	0.315
	Únstaged	8.92	0.06	1378.536	0.395	8.37	0.369	189.67	0.182	2.46	0.13	46.56	0.548
Localization	Lip	-	-	-	-	1	Reference			1	Reference		
	Cavity of the mouth	-	-	-	-	2.66	0.80	8.85	0.112	3.47	1.22	9.75	**0.019**
	Oropharynx	-	-	-	-	2.75	0.82	9.25	0.101	3.01	1.06	8.51	**0.038**
	Nasopharynx	-	-	-	-	0.99	0.18	5.48	0.986	2.47	0.72	8.40	0.149
	Hypopharynx	-	-	-	-	2.61	0.76	8.99	0.129	3.27	1.14	9.40	**0.028**
	Larynx	-	-	-	-	1.35	0.40	4.56	0.632	80	0.63	5.11	0.271
	Nose/paranasal sinus	-	-	-	-	2.01	0.468	8.81	0.355	2.0	0.60	6.90	0.258
	Middle ear	-	-	-	-	-	-	-	-	-	-	-	-
	Salivary glands	-	-	-	-	1.54	0.40	5.99	0.532	1.91	0.62	5.91	0.261
	Unspecified	-	-	-	-	-	-	-	-	-	-	-	-
Recurrence	No	1	Reference			1	Reference			1	Reference		
	Yes	0.11	0.02	0.77	**0.027**	0.12	0.05	0.29	**<0.001**	0.45	0.32	0.63	**<0.001**
Treatment	No therapy	1	Reference			1	Reference			1	Reference		
	Radiotherapy	0.10	0.05	0.23	**<0.001**	0.19	0.12	0.31	**<0.001**	0.37	0.25	0.533	**<0.001**
	Surgery	0.60	0.35	1.04	0.068	0.55	0.37	0.82	**<0.001**	0.51	0.36	0.73	**<0.001**
	Multimodal therapy	0.02	0.01	0.03	**<0.001**	0.05	0.04	0.08	**<0.001**	0.10	0.00	0.13	**<0.001**

* All significant variables (*p* < 0.05) of the univariate analyses were included in the multivariate analyses. All models included simultaneously the parameters gender, age, localization, recurrence, and treatment. To avoid a multiple inclusion of the same data, the models additionally included either the T, N, M classification parameters, or the UICC staging, or the SEER staging; OR = odds ratio; CI = confidence interval; - = not applicable; SEER = Surveillance, Epidemiology, and End Results Program; ** significant values (*p* < 0.05) in bold.

## Data Availability

The datasets used during the current study are available from the corresponding author upon reasonable request.

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
