# Peer review of "Early Mortality among Patients with Head and Neck Cancer Diagnosed in Thuringia, Germany, between 1996 and 2016—A Population-Based Study"

_cancers, 2022, doi:10.3390/cancers14133099_

Round 1

Reviewer 1 Report

This is a well written manuscripts on the early mortality of head and neck cancer patients on a  population-based cohort in Germany.

The results and conclusions are solid to arouse the interests among the readers and may thus generate a positive impacts on the clinical management of head and neck cancer patients.

Two minor suggestions:

1. The authors should clarify if the study exclude multiple head and neck cancer patients. 

2. The authors may consider further compare the difference on cause of death among 30, 60 and 60 days after diagnosis.

Reviewer 2 Report

Dear authors,

thank you for your interesting manuscript on analyses of data on 8288 patients with head and neck cancers diagnosed and reported to the Cancer Registry of the State of Thuringia between 1996 and 2016. 

During the review, I asked myself the following question: Why were disjoint subpopulations not compared e.g. those who died by day 30, those who died between day 31 and day 90, and those who died between day 91 and day 120? This approach might have yielded clearer results.

To achieve publication readiness, I recommend the following additions.

1. The title should indicate that the underlying population is the state of Thuringia.

2. The method for multivariate analyses needs a few more explanations in the methods section (e.g. type of regression model).

3. Table 1 should be significantly reduced in size. The columns with complementary numbers and percentages are not needed. Furthermore, the report on a single staging classification (e.g. UICC) would be sufficient. 

4. In the discussion of the limitations of the data and analyses, it should be mentioned that information on comorbidity is not available in the cancer registry, which presumably has a decisive influence on early mortality.

5. Since the legal regulations and responsibilities for cancer registration in Germany have changed within 20 years, bias and underreporting of incident cases cannot be excluded.

6. Likewise, it can be assumed that the modalities of therapy have changed during the same period.

7. Table 1 should be significantly reduced in size. The columns with complementary numbers and percentages are not needed. Furthermore, the report on a single staging classification (e.g. UICC) would be sufficient (Tab. 1 & Tab. 2). The relevant text passages would have to be revised accordingly.

8. Readability and clarity of text and tables should be improved if only two decimal places appeared for ORs and CIs. The p-values should remain with three decimal places.

9. The use of thousand separators should be consistently formated over the whole text.

Reviewer 3 Report

Interesting paper, well written. Only minor concernings:

- several typos should be corrected.

- Head and neck malignancies comprise a heterogeneous group of malignancies that cause significant morbidity to those affected. These malignancies are associated with specific risk factors and exposures, some of which impact prognosis. The most common risk factors for developing head and neck cancers are tobacco and alcohol use. Marijuana and e-cigarettes, occupational exposures, and use of topical substances have also been linked to head and neck cancers. Human papilloma virus has been associated with oropharyngeal cancer. Such measures as oral hygiene, screening, smoking cessation, and vaccination are measures taken to decrease the incidence and morbidity of head and neck cancers. please cite doi:10.1016/j.coms.2018.06.001

- The head and neck district represents one of the most common sites of onset of oncological diseases, with a high percentage of metastatic disseminations both locally and remotely. The prognosis of these tumors is closely linked to some main factors: a) the stage of the disease; b) loco-regional relapses; c) distant metastases. In head and neck cancers, distant metastases are present in about 10% of cases at the time of the first diagnosis and become evident in the course of the disease in a further 20% -30% of cases. When a distant metastasis originating from a head and neck tumor becomes evident, the prognosis is usually considered poor, with an estimated average survival of around 10 months. The aim of this work is to provide an updated and comprehensive update on the subject of distant metastases in cervico-cephalic oncology in the light of the most recent knowledge. Recently acquired concepts such as the molecular structure of tumors, the possible interactions between tumor cells and tissues, the peculiarities of oligometastatic disease, the role of immunotherapy are profoundly changing the therapeutic approach in these patients, with interesting repercussions in terms of control. of illness.. discuss and cite doi:10.14639/0392-100X-suppl.1-40-2020

- strobe guidelines should be applied to improve the overall quality

- discussion, chemoradiotherapy as a first-line therapy for head and neck carcinomas, although of choice for advanced stages, undergoes significant side effects with survival curves similar to surgery. please  doi:10.1016/j.anl.2021.05.007
